# Current Siberian heating is unprecedented during the past seven millennia

Rashit M. Hantemirov [1,2] ✉, Christophe Corona [3,4,5], Sébastien Guillet [3], Stepan G. Shiyatov[1,10], Markus Stoffel [3,5,6], Timothy J. Osborn [7], Thomas M. Melvin[7], Ludmila A. Gorlanova [1], Vladimir V. Kukarskih [1,2], Alexander Y. Surkov[1], Georg von Arx [8,9] & Patrick Fonti [8]

The Arctic is warming faster than any other region on Earth. Putting this rapid warming into perspective is challenging because instrumental records are often short or incomplete in polar regions and precisely-dated temperature proxies with high temporal resolution are largely lacking. Here, we provide this long-term perspective by reconstructing past summer temperature variability at Yamal Peninsula – a hotspot of recent warming – over the past 7638 years using annually resolved tree-ring records. We demonstrate that the recent anthropogenic warming interrupted a multi-millennial cooling trend. We find the industrial-era warming to be unprecedented in rate and to have elevated the summer temperature to levels above those reconstructed for the past seven millennia (in both 30-year mean and the frequency of extreme summers). This is undoubtedly of concern for the natural and human systems that are being impacted by climatic changes that lie outside the envelope of natural climatic variations for this region.

Whereas the globe is nowadays approximately 1.2 °C warmer than during pre-industrial times (1850–1900)[1,2], near-surface regions of the northern hemisphere high latitude have warmed at a rate nearly twice that of lower latitudes[1,3], a phenomenon known as Arctic Amplification[4]. The 2011–2020 mean temperature of the Arctic region (>60° latitude) alone was 0.71 °C higher than the preceding decade mean[5]. Adverse consequences of this rapid warming are already underway: enhanced ice loss across Greenland[6], record-low Arctic sea ice extent[7], permafrost thawing[8,9] and unprecedented wildfires across Siberia[10,11]; with considerable implications on a suite of human and natural systems, both within and outside the polar regions[4]. Yet, despite the relevance of temperature for the Arctic system, instrumental records are often short or incomplete in polar regions[6] and

precisely-dated temperature proxies with high temporal resolution are largely lacking[12].

Siberia is among the regions with the strongest warming worldwide (Fig. 1a) and heatwaves have reached a disturbing new level in recent years, especially in 2020 when temperatures soared across Siberia to reach a record-breaking 38 °C inside the Arctic Circle[3,13]. This massive build-up of heat in the Arctic promotes the rapid disappearance of sea ice, accelerates thawing of carbon-rich permafrost[14–16] and the emergence of extreme wildfires[11]. A hotter climate in Siberia arguably has devastating, cascading effects on local ecosystems, human communities, and the built environment[17–19]. It will also impact the global climate system as a whole, for example through the enhanced release of greenhouse gases from permafrost bodies[20],

[1]Institute of Plant and Animal Ecology, Ural Division of the Russian Academy of Sciences, Ekaterinburg 620144, Russia. [2]Ural Federal University, Ekaterinburg 620002, Russia. [3]Climate Change Impacts and Risks in the Anthropocene (C-CIA), Institute for Environmental Sciences, University of Geneva, 1205 Geneva, Switzerland. [4]Geolab, UMR 6042 CNRS, Université Clermont Auvergne, F-63057 Clermont-Ferrand, France. [5]Department F.A. Forel for Environmental and Aquatic Sciences, University of Geneva, 1205 Geneva, Switzerland. [6]Department of Earth Sciences, University of Geneva, 1205 Geneva, Switzerland. [7]Climatic Research Unit, School of Environmental Sciences, University of East Anglia, Norwich NR4 7TJ, UK. [8]Swiss Federal Research Institute WSL, 8903 Birmensdorf, Switzerland. [9]Oeschger Centre for Climate Change Research, University of Bern, 3012 Bern, Switzerland. [10]Deceased: Stepan G. Shiyatov. ✉e-mail: rashit@ipae.uran.ru

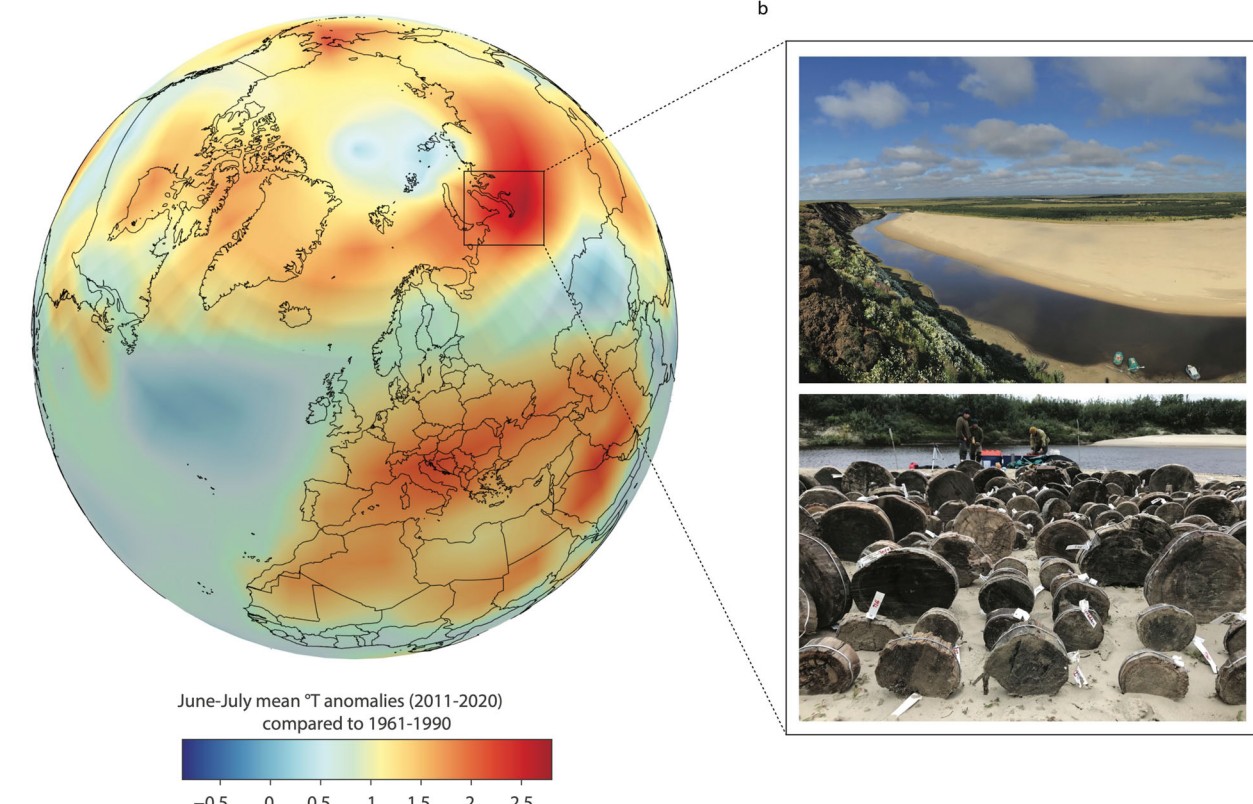

**Fig. 1 | Study region and sampled material. a** Mean June–July temperature anomalies over the Northern Hemisphere and the larger Yamal Peninsula region (black rectangle) over the last decade on record (2011–2020). Temperatures are expressed as anomalies relative to a 1961–1990 baseline climate using the HadCRUT.5 dataset[5]. **b** Aerial view of one of the sites sampled, i.e. Tanlova River, Yamal Peninsula and the subfossil wood collected during the most recent field campaign in August 2019.

the largest peatland surface on Earth[21], and wildfires[10], reduced snow cover and melting ice, or even changes in the physical processes of the Arctic Ocean[22].

High-resolution Holocene proxy records are critically lacking in Siberia[23,24], rendering efforts to place the ongoing warming into a longer-term perspective very challenging. Many existing Arctic proxy records are from North America, Scandinavia and Greenland, whereas the Siberian Arctic remains critically underrepresented in natural archive compilations to the present day[25,26]. Moreover, most existing Arctic proxies suffer from limited temporal resolution and limited coverage of the most recent decades. While providing insights into longer-term Holocene changes, proxy archives from the Arctic have hitherto remained largely restricted to low-resolution pollen assemblages and laminated lake sediments. Though these records preserve valuable information on climate variability for periods exceeding 2000 years, they retain virtually no variability at timescales shorter than 300 years[27]. Also, most proxy records end in the mid-twentieth century and therefore cannot provide lessons for the most recent past and the ongoing, accelerated climate warming. As a result of the limited temporal resolution of these proxies, the heterogeneity of Arctic climate[28], the spatial interpolations used and the missing coverage of the recent warming, no robust consensus has hitherto been reached on the overall rate and magnitude of industrial-era warming in a Holocene context for the Siberian Arctic[27,29,30]. This lack of consensus has also hampered the use of analogs from the distant past to anticipate changes in a future, warmer Arctic[29].

Here we use annually resolved, summer temperature sensitive tree-ring records from Yamal Peninsula (67–67.8 °N; 69–71 °E; NW Siberia, Russian Federation) to reconstruct 7638 yrs of Siberian Arctic summer temperature variability at interannual to centennial timescales (5618 BCE to 2019 CE). Thanks to this exceptionally well-replicated and well-balanced multi-millennial long tree-ring width chronology, we are able to provide a highly-resolved robust reference to benchmark the recent warming over a significant large proportion of Siberia. The reconstruction indicates the occurrence of a strong warming trend after the culmination of a multi-millennial Holocene summer cooling trend at the end of the Little Ice Age. Our analysis demonstrates that the rate of industrial-era warming and the magnitude of the temperature anomalies in recent decades are unprecedented for the mid and late Holocene in the Yamal region.

## Results and discussions
### The *Yamal7k* tree-ring chronology

Yamal—meaning *"the end of the land"* in the indigenous Nenets language—stands among the northernmost regions containing living forests and myriads of subfossil trees conserved in the riverbanks (Fig. 1b). Although forests are nowadays mostly absent in the Yamal region – with the exception of the southern part of the peninsula —we can rely on remnants of past forests that penetrated deep into the tundra along the river valleys as a major source of subfossil wood. Trees exposed to the ever-active eroding rivers were trapped in situ for centuries to millennia in old meanders and are now released from river sediments by erosion. The Yamal tree-ring chronology has been developed by collecting these subfossil trees over the past 40 yrs during extensive fieldwork, making it the longest and best replicated tree-ring width (TRW) record from the Arctic with more than 4800 collected samples[31]. The most widespread tree species in the collection is *Larix sibirica* Ledeb. representing more than 91% of all samples. Other represented species are *Picea obovata* Ledeb (6%) and *Betula pubescens* ssp. *tortuosa* (Ledeb) (3%).

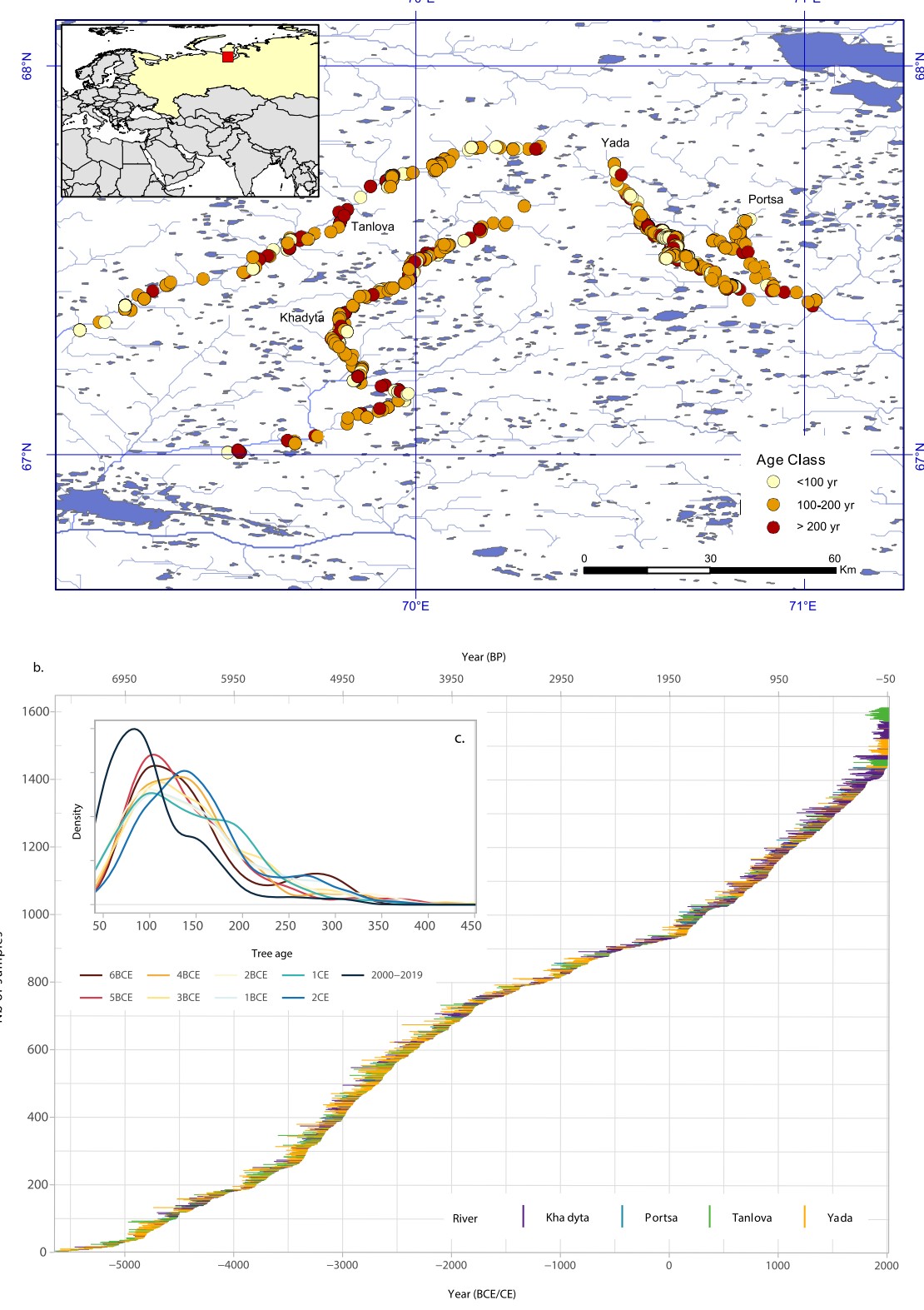

**Fig. 2 | Sampling locations and temporal coverage of the Yamal7k tree-ring chronology. a** Sampling locations and age class of the Yamal7k tree-ring chronology with indications of the four low-flow rivers. **b** Temporal distribution of 1611 individual series (1425 subfossil logs and 186 increment cores from living trees) color coded by the corresponding river. The mean segment length is 142 years (ranging from 41 to 452). **c** Ring age distribution across millennia.

The dataset used in this study contains 1611 TRW series (i.e. 1425 subfossil logs and 186 increment cores from living trees) from *L. sibirica* trees collected along four south-flowing rivers at elevations ranging from 20 to 50 m asl[32]. The number of replicated series for each year ranges between 4 and 187, with an average replication of 30 series (Fig. 2).

The sampling region is located in the permafrost zone where seasonal melt reaches depths of up to 2 m. The region is also known as a hotspot of surface air temperature warming that can be ascribed in

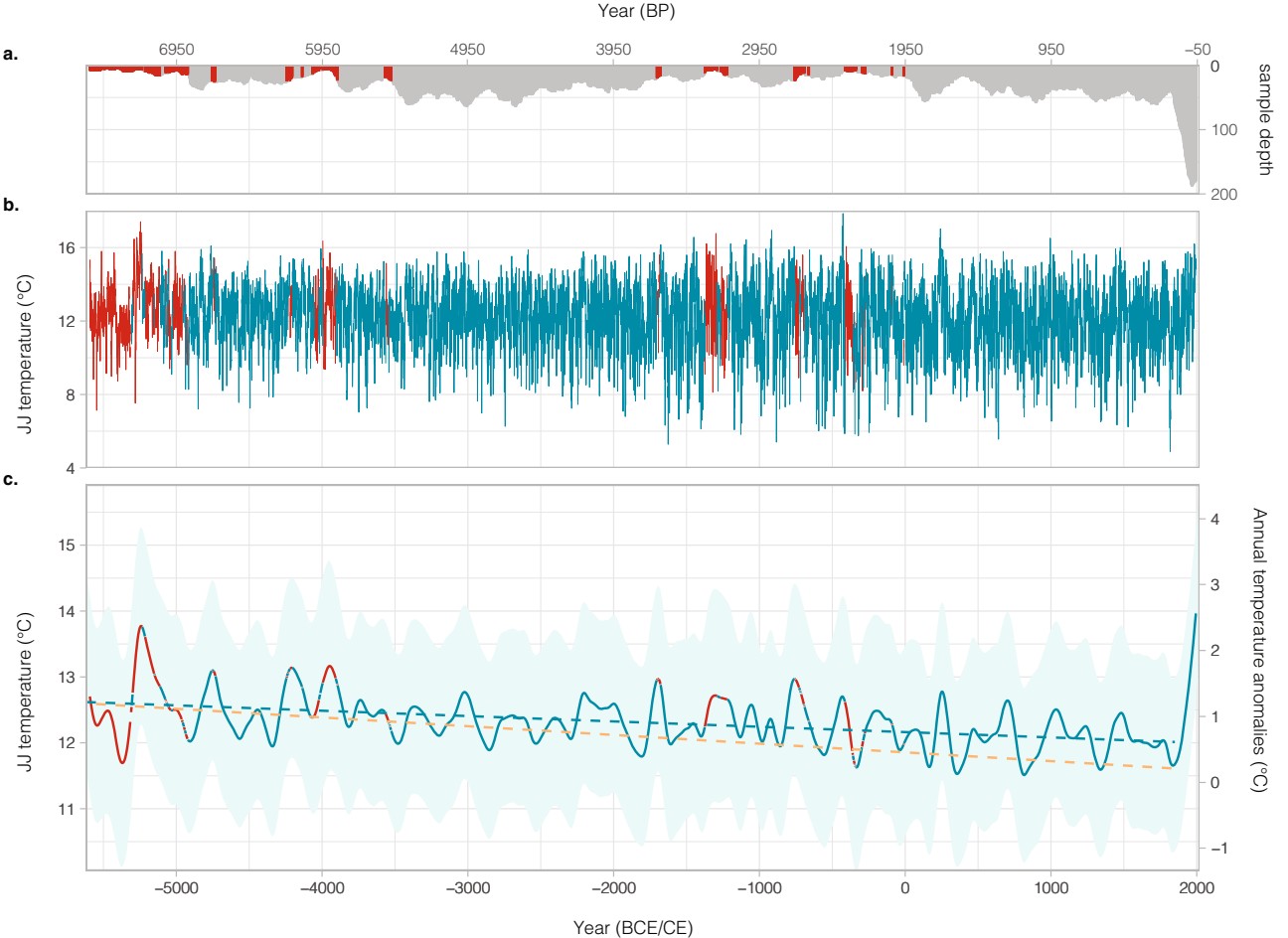

**Fig. 3 | Reconstructed Yamal June–July (JJ) temperature over the past 7638 years. a** Sample depth for each year over the past 7638 years. The sample depth is constantly ≥ 4 since -5618 BCE with an average of 30 samples covering each individual year (with a range from 4 to 187 samples). **b** Annually resolved June–July temperature reconstruction at Yamal Peninsula and **c** 200-year cubic spline-smoothed chronology showing a distinct Holocene cooling trend until the end of the Little Ice Age around 1850 and a sharp industrial warming over the past 170 years. The magnitude of the long-term cooling trend is insensitive to the choice of start and end years of the trend. The reconstruction uncertainty estimates (light blue band) incorporate both the ring-width chronology error and the reconstruction error (see Material and Methods). Red color indicates periods when Expressed Population Signal (EPS) is below 0.85. The blue and dashed orange regression lines show the Holocene cooling trends (over 5618 BCE to 1850 CE) from the Yamal7k JJ reconstruction and from the 60–90° N reconstructed annual temperature anomalies of the 12k Database[23] and referenced to the 1800-1900 AD mean (right-hand scale), respectively. The latter is a multi-proxy reconstruction (mostly pollen assemblages, chironomids and marine sediment cores) of annual temperature with temporal resolutions at centennial scale.

part to the amplified warming of the Barents and Kara Seas[33]. With current (1991–2020 and 2011–2020) June–July (JJ) warming at +1.32 °C and +2.33 °C, respectively, compared to the climate norm 1961–1990, or even +2.02 °C and +3.06 °C, respectively, compared to pre-industrial levels (1850–1900; ref. 5), Yamal Peninsula shows the largest JJ warming over land across the Arctic regions (Fig. 1a).

**A robust reconstruction of Siberian summer temperatures**

To preserve long-term climatic variability in the TRW chronology while removing artefacts related to the age of individual tree rings, we applied a multi-curve signal-free Regional Curve Standardization (RCS) using two age-aligned growth-rate curves computed for fast- and slow-growing trees. This standardization has proven useful to circumvent different average growth rates between subfossil and contemporary material and to avoid a possible inflation of recent chronology values arising out of inadvertent selection of relatively fast-growing trees in recent centuries, a problem referred to as the "modern sample bias"[34] (see "Methods", and Supplementary Fig. 1).

The Yamal7k TRW record is expected a priori to provide a very good estimate of past summer temperatures because it is highly replicated and the samples were taken at the far northern tree line

where growth is strongly limited by summer air temperatures. This dominant limitation on growth strengthens the correlation between individual trees (Rbar ranging from 0.27 to 0.80 and averaging 0.53, Supplementary Fig. 2), enabling individual rings to be cross-dated with high accuracy and increasing confidence in the mean chronology (as measured by the RCS-adjusted EPS, Supplementary Fig. 2), as well as maximizing the sensitivity to a single climate variable, namely JJ temperatures. We confirmed this process-based understanding and identified the optimal seasonal sensitivity by correlating the Yamal7k chronology with sets of 5-day (pentad) mean air temperatures from the Salekhard meteorological station over the period 1883–2019 (Supplementary Fig. 3). The optimal calibration window was defined by computing correlations between the 2-curve SF-RCS chronology and sets of 5-day (pentad) mean temperatures from the Salekhard meteorological station over the period 1883–2019 (Supplementary Fig. 3). The Yamal7k chronology explains a large portion of significant JJ temperature variability over most of Siberia. The highest spatial field correlation ($r > 0.6$) covers a region comprised between 16° and 18° E and above 60° N (Supplementary Fig. 3c). Four reconstructions were realized and calibrated against a 50-day mean summer temperature window (June 16 to August 4, $r = 0.75$, $p < 0.001$) using ordinary least

squares regression, variance scaling, a multi-frequency band approach, and K-fold mean (see "Methods", Supplementary Fig. 4). Split period calibration/verification statistics provide evidence that the Yamal7k record of relative tree growth yields a very good estimate of JJ temperature (Supplementary Table 1). As all four reconstructions are significantly correlated ($r > 0.97$, $p < 0.001$) among each other, we only analyze the one based on variance scaling (VS) hereafter and refer to it as *Yamal7k* (Fig. 3 and Supplementary Fig. 5).

Comparable strength of the relationship with air temperature ($r = 0.75$; independent verification RE = 0.70, CE = 0.42 for 1883–1951 and RE = 0.67, CE = 0.36 for 1952–2019; see "Methods" and Supplementary Table 1) has not hitherto been achieved in multi-millennial TRW records[35], only in maximum latewood density (MXD) or wood anatomical reconstructions[36]. Likewise, the Yamal7k reconstruction also reproduces decadal JJ temperature variability (Fig. 3 and Supplementary Fig. 4) as attested by the strong correlation ($r = 0.92$; $p < 0.05$) between the 30-yr spline-smoothed chronology and meteorological records. Consistent with other observations in Siberia[37], Yamal7k also does not show any obvious signs of divergence between TRW and instrumental temperatures (Supplementary Fig. 4), unlike various other records from high-latitude and/or high-altitude sites[38]. Moreover, the JJ air temperature fluctuation reconstructed from a resampled chronology designed to remove the higher replication over the modern period remain significantly correlated with the original VS reconstruction ($r = 0.986$, Supplementary Fig. 6), thereby confirming the robustness of the two-curve RCS method.

## Unprecedented warming

Over the period 5618 BCE–1850 CE, Yamal7k exhibits a clear and long-term cooling trend of $-0.08\,°C$ kyr$^{-1}$ (i.e. 0.6 °C; Fig. 3). This is in contrast to the majority of TRW records from the exhaustive PAGES2k dataset that do not show multi-millennial trends[39]. We tend to ascribe this ability to the high replication and even distribution of samples over time in Yamal7k that facilitate the application of an RCS standardization aimed at preserving long-term trends[39]. The Yamal7k cooling trend is significantly lower than the 0.31 °C kyr$^{-1}$ reported for maximum latewood density records in Norther Scandinavia over the last 2 millennia[40]. It is yet comparable to a recent composite of Holocene temperature reconstructions derived from lower-resolution proxies including ice cores, pollen, lacustrine and marine sediments (estimated to a median of 0.9 °C cooling across 60–90° N over the same period[41]). This long-term cooling observed in lower-resolution proxies[24,26] has been ascribed to orbital forcing of high-latitude summer insolation after the end of the Holocene thermal maximum[24,42], though it may not be present in all regions[30]. Yet, the orbital cooling trend at Yamal Peninsula was halted abruptly by pronounced industrial-era warming (1850–2019), leading to an increase of JJ air temperatures without any parallel for comparable time windows (i.e. 170 years) over the mid-to-late Holocene (Fig. 4a and Supplementary Fig. 7). Even for shorter time windows (between 37 to 170 years ending in 2019), mean temperatures are beyond the values reconstructed for any other period of the mid-to-late Holocene (Supplementary Fig. 8b). For windows 30 years and longer, the mean temperature of one of the periods after 2013 is always ranked nr 1 (Supplementary Fig. 8c), thereby underlining the exceptional character of the ongoing climate warming. The annual resolution of the Yamal7k record also allows a demonstration that climate norms calculated over 30-year periods ending at or after 2002 CE have started to exceed the range of reconstructed climate variability (>99 percentile) during the mid-to-late Holocene (Supplementary Fig. 8d).

Likewise, we observe an increase in extremely high ring-width indices during the last 100 years, from which we infer an increase in extremely warm JJ temperatures at Yamal (>95th percentile; Fig. 4d). Comparison of the distribution of annual air temperatures between 5618 BCE–1850 CE and 1851–2019 CE in fact highlights that the median

and 95th percentile of industrial-era temperatures have increased by 0.53 °C (from 12.43 to 12.96 °C) and 0.63 °C (from 14.80 to 15.43 °C), respectively (Supplementary Fig. 8). Figure 4d underlines the exceptional increase in the frequency of warm extremes over the past 100 years (1920–2019) with more than one in four years ($n = 27$) exceeding the 95th percentile of the distribution of reconstructed temperatures; of these, 19 occur after 1980 CE. At the same time, only three out of the last 100 years (1947, 1932 and 1971 CE) were below the 10th percentile of the coldest years, and not a single year falls within the 5th percentile of the coldest years reconstructed over the past 7638 years.

The high replication of annually resolved TRW records covering the past 7638 years and the use of state-of-art reconstruction procedures has, for the first time in the Siberian Arctic, enabled unraveling of both the Holocene summer cooling trend—reaching its culmination at the end of the Little Ice Age—and the inversion to an unusually strong warming trend. We demonstrate that the rate of industrial-era warming and the magnitude of the temperature anomalies in recent decades are unprecedented for the mid and late Holocene in the Yamal region. The mean temperature reconstructed over the period 1920–2019 (13.47 °C) very clearly exceeds the range of natural variability with a return period >4850 years based on a Generalized Pareto distribution fitted above the 50th percentile of the entire time series (Supplementary Fig. 9a, c). If the last 100 years are excluded when fitting the distribution, the estimated return period for the 1920–2019 mean air temperature suggests that the warmth of the last 100 years would have been virtually impossible at any time in the last seven millennia and in the absence of climate change. As such, both the mean JJ air temperatures (12.83 °C) and the warming rate (0.0173 °C yr$^{-1}$) observed between 1850 and 2019 exceed the range of natural climate variability in the larger study region (Supplementary Fig. 9c). Although our analysis does not include projections into the future, it cannot be ruled out that the rapid warming that we observe in our reconstruction and that is supported by multiple lines of observational evidence on the ground, may result in a new climate state in which heatwaves as well as the associated melting of permafrost bodies and occurrence of wildfires may become routine[3,7]. Given the damaging consequences from such climate dynamics[43], our results also point to serious risks of simultaneous adverse impacts over large areas if no adaptation strategies are adopted.

## Methods

### Dendrochronological data

We used increment cores from living trees ($n = 186$) and subfossil stem cross sections ($n = 1425$) of *L. sibirica* Ledeb. from the southern part of Yamal Peninsula (67.0° to 67.8° N), North-West Siberia (Russian Federation) to construct a tree-ring width (TRW) chronology built from at least 4 samples covering the period 5618 BCE to 2019 CE continuously (Fig. 2). The sampling area is within a huge old delta with south-flowing natural rivers with strongly meandering channels in shallow and wide valleys that are unearthing sandy permafrost soils with subfossil logs. Sample collection started in 1982 and 20 field expeditions have been organized since. Subfossil wood samples were collected in a responsible manner, with permission from local authorities and in accordance with local laws. Wood cores from living trees were taken at different locations from the currently forested areas (up to 67.5° N) next to the rivers.

If the root collar was visible (which was the case in 40% of the subfossil sampling), increment cores or cross sections were taken at heights ranging from 0.3 to 1.3 m, otherwise the sampling occurred at the lowest height of the log. From the more than 3500 available TRW series of subfossil larches, spruces and birches, approx. 1900 were cross-dated and, of these, 1425 subfossil larch series were included in the chronology. Ring width series of spruces and birches, as well as larches north of 68° N were not included in the chronology. The Yamal7k chronology is unusually well replicated with 30 samples being, on average, available for any given year;

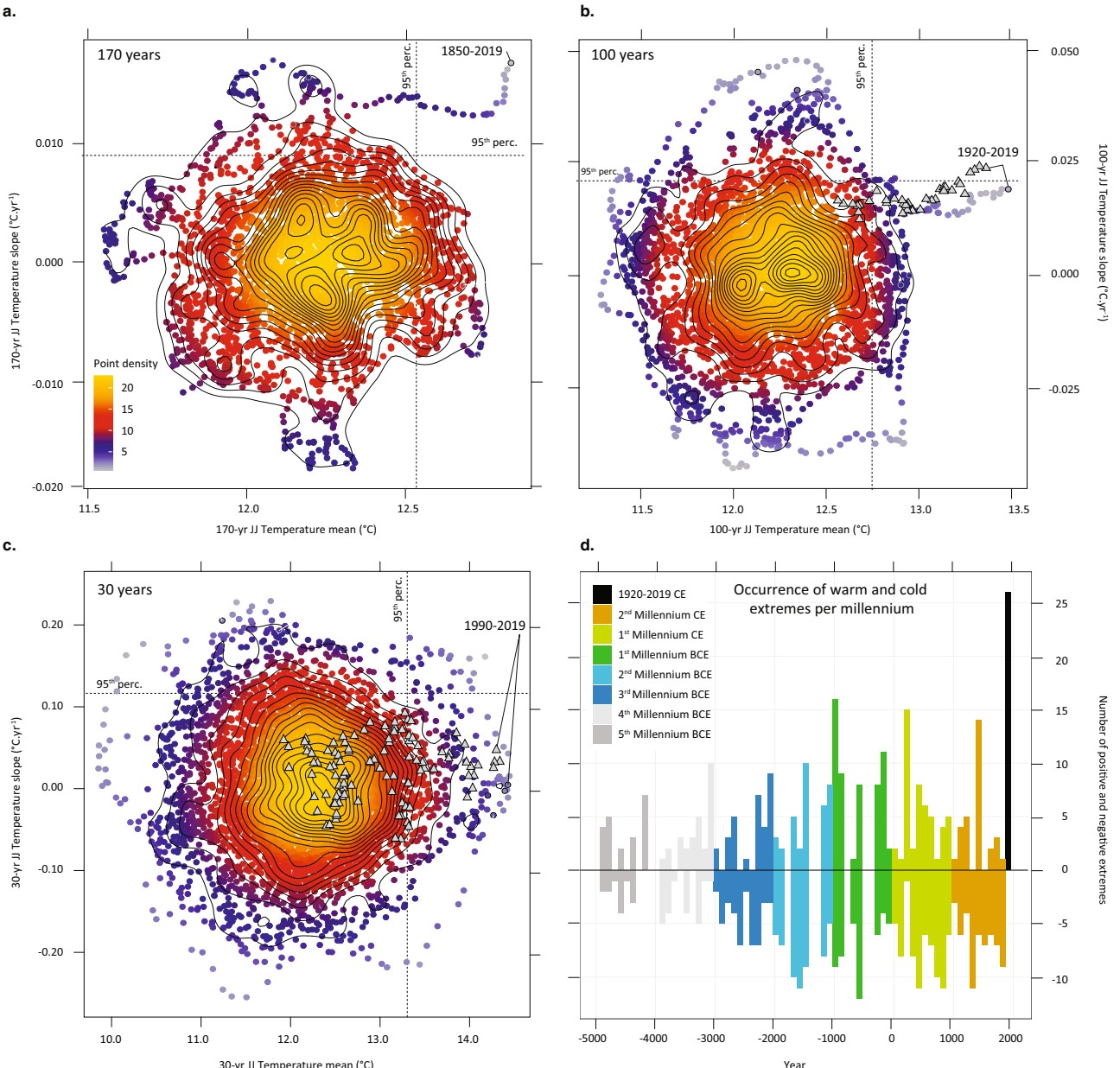

**Fig. 4 | Recent June–July (JJ) warming unprecedented over past 7638 years. a–c** Two-dimensional density plots of temperature trends versus mean temperatures computed for time windows of 170, 100, and 30 years. Gray triangles indicate mean temperature anomalies and slopes computed for the Salekhard meteorological dataset. Time windows including years with EPS < 0.85 were excluded from the analyses. Labeled dots indicate the period ending in 2019 CE. Results show that **a** industrial-era warming (1850–2019 CE) is unprecedented in terms of slope and magnitude, **b** mean JJ temperatures of the past 100 years (1920–2019) are warmest over the mid-to-late Holocene, and that **c** the 1990–2019 CE temperature norm exceeds all 30-year norms calculated over the past 7638 years, with values being out of range of reconstructed climate variability (>99 percentile) since 2002 (see also Supplementary Fig. 8d). **d** Frequency of hot extremes per century. With 27 warm extremes and a complete loss of cold extremes, the past 100 years were beyond the 95th and the 5th percentiles, respectively.

sample depth exceeds 5 trees in 98.6% of the chronology (i.e. 7528 out of 7638 years) (Fig. 2).

**Chronology development and temperature reconstruction**
We applied a two-curve signal-free regional curve standardization (2-curve SF-RCS[44,45]) as it allows the removal of age-related trends while preserving low-frequency variability in tree-ring data and is particularly suitable where large amounts of subfossil series are available, such as in this study (Supplementary Fig. 1). To sort trees by "growth rate", we compared each tree's growth rate with the growth of a single RCS curve built from all available trees. The relative growth rate was obtained by the ratio between the sum of measurements of a tree (i.e. the radius of the tree) divided by the sum of the RCS curve values

derived from all trees. Trees were then sorted by "relative" growth rates and equal numbers of trees were used to create series of RCS curves representing increasing growth rate classes[46]. The Expressed Population Signal (RCS-adjusted EPS, i.e. adjusted to also represent long-timescale uncertainty[34,45]) was used to determine whether the TRW chronology has a clear population signal, which is necessary to obtain a reliable climate reconstruction[47] (Supplementary Fig. 2). RCS-adjusted EPS exceeds the 0.85 threshold over 87.1% of the chronology while the average Rbar calculated in 50-yr windows is 0.53 (ranging from 0.27 to 0.80).

The 2-curve SF-RCS Yamal chronology was correlated with daily air temperatures from the Salekhard meteorological station (66°32′ N, 66°32′ E; 35 m a.s.l.) continuously available since 1883 CE

(Supplementary Fig. 3). We tested correlations for different periods ranging from 5 days to 4 months. Correlations between measured air temperatures and the TRW chronology are highest ($r > 0.75$) for windows starting on June 16 and ending on July 20 (that is 7 pentads) to August 4 (10 pentads) (Supplementary Fig. 2). In this study, we selected the June 16 – August 4 windows as it integrates the longest time window. This empirical seasonal sensitivity is supported by weekly observations of tree-ring formation performed in the vicinity of Salekhard station during the boreal 2018 and 2019 growing seasons (see Supplementary Figs. 10–12) and therefore used for the Yamal7k temperature reconstructions. Three different reconstructions were obtained, namely the ordinary least square regression (OLS)[48], the variance scaling (VS)[48] and the multiple frequency band (MFB)[34,49] approaches. A split period (1883–1951, 1952–2019) calibration/verification procedure was then applied to assess the skills of the reconstructions, whereas the final models were built over the entire 1883–2019 period (Supplementary Table 1). To test for potential biases induced by increasing replication over the modern period, we subsampled the Yamal7k chronology to adjust sample depth (with a mean of 35 samples over the period 1800–2019 CE) to a level that is comparable to that over the last seven millennia (with a mean of 30 samples). The two VS reconstructions computed from the original and resampled Yamal7k chronologies are presented in Supplementary Fig. 6. In addition to the commonly used calibration-verification method, the K-fold cross-validation[50] method was also used to test the reconstruction function to exclude possible biases related to the progressive warming in more recent years. Statistics used to evaluate the quality of the estimates included $R^2$, $R^2$ adjusted, RE, CE, and full period Durbin–Watson (range) statistics (Supplementary Table 1). Because all reconstructions were significantly correlated ($r > 0.97$, $p < 0.001$) among each other, we only analyzed the reconstruction based on variance scaling (VS). For the variance scaling approach, uncertainties have been computed following Briffa et al.[34]. For each year, the reconstruction uncertainty is the square root of the sum of two squared uncertainty components: the ring-width chronology error and the temperature reconstruction error. The time-varying ring-width chronology error represents the chronology standard error (i.e. the standard deviation of all tree index values in each year divided by the square root of the number of trees in that year) scaled (multiplied) by the same scaling factor used to produce the reconstruction. The time-invariant temperature reconstruction error is equal to the Standard Deviation of the residuals (i.e. instrumental temperature minus estimated values) over the calibration period.

### Quantification of return periods

To quantify the return period of the air temperatures observed over the 1980-2019, 1920–2019 and 1850–2019 CE periods, we fitted generalized Pareto Distributions (GPD) to reconstructed June–July (JJ) average temperatures and slope time series computed over time windows of 30, 100, and 170 years, respectively, shifted by 1–170 years. In the case of all series, we both included and excluded the reference time periods 1980–2019, 1920–2019 and 1850–2019 CE, and show that the GPD is fitted to data points that are in excess of suitably chosen thresholds. We used standard tools for threshold selection including the mean residual life (MRL) plot and threshold stability plots with the *fevd* routine from the *extRemes* package R. Results are summarized in Supplementary Fig. 9.

## Data availability

The data used to perform our analysis as well as our results have been uploaded to Zenodo and are accessible using the following link https://doi.org/10.5281/zenodo.6477133.

## Code availability

The codes that support the findings of this study are available from the corresponding authors upon reasonable request.

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

## Acknowledgements
R.M.H., S.G.S., A.Y.S., and L.A.G. received funding from the Russian Foundation for Basic Research (no. 18-05-00575). M.S., C.C., S.G., and P.F. received funding from the SNF Sinergia project CALDERA (no. 183571). V.V.K. acknowledges support from the Russian Science Foundation (no. 21-14-00330). G.vA. acknowledges support from the SNF project XELLCLIM (no. 182398). T.J.O. acknowledges support from UK NERC project GloSAT (no. NE/S015582/1).

## Author contributions
S.G.S. conceived the study. R.M.H. and S.G.S. with input from V.V.K. and A.Y.S. organized the sampling. The tree-ring width measurements and the cross-dating were performed by R.M.H., S.G.S., A.Y.S., and L.A.G.; T.M.M., T.J.O., and R.M.H. performed the chronology development and reconstruction with input from S.G., C.C., M.S. and P.F. Quantifications of extremes and return period were performed by C.C., S.G., M.S., R.M.H., G.vA., and P.F. The paper was written by C.C., S.G., M.S., R.M.H., T.J.O., and P.F.

## Competing interests
The authors declare no competing interests.
