## [Peer Review File · Nature Communications]

Current Siberian heating is unprecedented during the past seven millenniaReviewers' Comments:

Reviewer #1:

Remarks to the Author:

The authors provided well-replicated, high-quality 7,638 years-long tree-ring width chronology at Yamal Peninsula that is one of the northernmost forested region in the World. The chronology is very sensitive to June-July temperature variations. This annually-resolved data set is an impressive paleoclimatic archive that is used to assess the rates and scale of the modern temperature changes in the Arctic in comparison with those over the mid and late Holocene. The chronology itself is the main achievement of this work and this archive will be certainly used in the future by hundreds of researches. The authors identified a weak (0.6C) long-term negative linear trend in the June-July temperature over the entire period covered by their data and claim that this trend was reversed in the industrial era. They consider the current JJ temperature as unprecedented in comparison with the pre-industrial time (from the 8th millennium to 1850s AD) in many respects, namely in term of the number of positive extremes and the degree of current warming.

The manuscript is a result of the efforts of a large group of researchers that lasted for several decades. The chronology is exceptional: it is well-replicated, sensitive, very solid. It is constructed from the wood collected from the sediments of the rivers flowing south bringing from the north more temperature sensitive samples.

The applied methodology is correct; the work meets all international tree-ring standards. This is clearly a precious contribution to the international paleoclimatic activity and, in my opinion, after clarification of some details the paper should be published in the Nature Communications.

Below are some comments that I would like to address to the authors and the Editor

1. Long-term trends

It is difficult to compare the 8 ka trend with the one a century-long. Looking at the Fig. 2 the reverse is not obvious, probably due to the scale of the graph. In addition, one can find – at least at the graph – other cases when the long-term trend was reversed for several decades. While there are no doubts that the inter-annual variations are well preserved in the chronology the longer-term variability in tree ring records is often questionable. The authors used the multi-curve signal-free Regional Curve Standardization (RCS) in order to preserve as much as possible this low-frequency signal. Moreover, they used two growth-rate curves for fast- and slow-growing trees. This approach is correct, and nowadays I do not know a better way to deal with this problem. To secure their conclusions they also compare their records with the trend from the lower resolution proxies and find them to be similar (0.9 versus 0.6 C). The authors admit that "This is in contrast to some other TRW records that do not show multi-millennial trends". I would suggest to go into details here, to give appropriate references and explore potential differences in approaches and provide more solid evidence that this trend is not a statistical artefact. What is a difference in methodology and data set in this paper in comparison with others that do not reproduce the long-term trend? Is the 0.3C difference negligible? Do I understand correctly that the mean temperature at the Fig. 2 (white dashed line for the area across 60–90°N) is up to 0.5 C higher than the mean in Yamal that is located at 67 N, i.e. at the south of the area?

2. Current warming

The tree-ring-climate model seems to be very solid – the meteorological records are long, the correlation is high. However, at the Fig 2 only very few last years look anomalous especially when averaged, the unsmoothed values do not look unusual. Could it not be partly just a statistical edge effect of the smoothing at the end of the series?

When I look at the Fig. 1 in Mann, 2021 (Mann M. E. Beyond the hockey stick: Climate lessons from the Common Era //Proceedings of the National Academy of Sciences. – 2021. – T. 118. – №. 39) I see a clear picture of the unusual warming; when I compare it with the Fig. 2 in this paper I begin to hesitate. Probably it is a matter of representation of the results, but it might be useful to think about the strengthening of the visual message.

The sample population of 2000-2019 AD is different from those of the other millennia – it is larger and

the trees are younger. At the ED Fig. 2 it is shown that the extreme growth at the end of the curve is rather sensitive to the way you choose for standardization.

The ED Fig. 2 shows that there are substantial differences in the growth of trees younger than and older than 100. Can you show a comparison of "the young" trees and "old" trees chronologies with the Salekhard temperature to see how well it is reproduced by the two subsets?

Did the CO₂ fertilization contribute the growth in 20th-21th centuries apart from the climate warming?

3. Other warmings

What are three other warmest (high number of extremes) centuries at the Fig 3? Can you zoom up somehow these periods to show that they are different from 1920-2019 AD? At the Fig. 3 very interesting "prominences" pop out of the point cloud. Each "prominence" corresponds to some anomalous period; it is interesting to see what these periods are and how they are formed. The problem of the scale of the warming during the Medieval Anomaly is still under discussion. It would be interesting to show more explicitly this comparison. Fig. 3 shows a very interesting pattern of smaller number of extremes between -2000 to -4000 years. Is it an artefact or a real climatic pattern?

Reviewer #2:

Remarks to the Author:

The manuscript entitled "Current Siberian winter is unprecedented during the last seven Millennia" by Hantemirov et al. presents a novel and multi-millennial tree-ring width chronology from the Siberian Arctic region using three different tree species. The displayed results are based upon a 7,623 year spanning chronology, built from recent as well as historic samples. The displayed results are in general sound, sufficiently well-explained and presented through well—designed figures as well as by underlying statistical analyses. The grammar as well as the overall language is easy to read, while using the respective necessary scientific terms and language. The presented results steadily support the hypotheses/ intended scientific gaps for the region. The used literature is state-of-the art, although the authors could have used more recent multi-millennial work from other authors e.g. for the comparison of their new data with other time series (see my specific comments on that below). From my perspective, I have not read such a nicely presented and sound manuscript for months.

The authors present their results throughout the manuscript in a clear and sound way. The performed analyses are comprehensible, use state-of-the art dendroclimatological methods and are illustrated in an impressive way. However, some questions arise while reading through the manuscript that justify my review statement as minor revisions.

- Why did the authors use the climate normal period from 1961-1990 instead of using the more recent normal period? At least a comprehensive comparison of both would be helpful to illustrate readers beyond dendroclimatology possible impacts caused by anthropogenic warming for the resulting reconstruction. This also holds true for the main text comparisons (e.g. line 82).

- Please describe in more detail the choice of the split sample calibration for your chronology in the methods. This should include some explanation why you neglect or dismiss other state-of-the art methods (e.g. LOOC) or more novel approaches (e.g. K-folded mean) in dendroclimatology. This is especially of importance for the K-folded mean method, since the recent period (1952-2019) is within the most amplified warming period in the used climate record.

- In addition to this, have you checked the usage of gridded re-analysis products to further underline your results? How about analyses investigating the regional representativity of your reconstruction? This could stimulate the discussion part and the outreach points on the overall usage of your dataset e.g. compared to datasets from High Asia (e.g. by the tree-ring oxygen isotope based dataset presented in Bao et al, PNAS 2021)

- The overall comparison with other regional / northern hemispheric temperature proxy records represented in the ms is astonishingly low and deserves more focus. You mentioned of course the lack of available high-res (TRW) data, but there are records from other proxy archives and / or from other Arctic locations in the Northern hemisphere that are worth for comparing. This includes incorporating a graph illustrating common periods or contrasting trends also used for the overall discussion (e.g. with the recent NH recon comparison paper by Buntgen et.al. 2021). How about using additionally modelled /simulated GCM paleodata for comparison?

More specific comments:

- Please insert in Fig 2 the running sample replication and the EPS as individual layers. It is only readable if using a magnifier when the individual series shifts from grey to purple... This clearly enhances the overall readability/information on time series statistics. The individual series given in grey can be then e.g. in a more light grey.

- Extended data Fig. 1.: the periods with insufficient sample replication should be marked more prominent. Within the used coloring scheme, it is not easy to recognize that. In addition, you stated the "Sentinel scene is used for giving an overview of sample replication. Well, I have never heard that a satellite image can be used to get information on sample replication, I think you meant that you used it for illustrate a) location of samples in your study area and b) additionally give information on the respective location of long/short series. However, the overall readability of this in your graph a) is weak. Why did you not use simply different age classes instead of a floating classification? It is quite hard to read anything in the color floating legend where e.g. the more older samples are from (tiny graph inlay for the amount of information you present here).

Reviewer #3:

Remarks to the Author:

Peer Review

RE: Current Siberian heating is unprecedented during the last seven millennia

Authors: Rashit M. Hantemirov et al. 2021

Nature Communications manuscript: NCOMMS-21-43050-T

Preface:

To the editor, this is not an anonymous review. Please share the entirety with the authors.

Summary:

I am impressed. This work clearly represents a tremendous effort spanning many years and involving many participants on multiple levels. I am very pleased to see it finally coming together in a such a well presented and thorough manner. It had been a long time coming, a chronology of this length from a region of significant paleoclimate importance. As mentioned, the manuscript is well constructed, written, and cogent. I have found absolutely no glaring technical issues regarding the analyses involved to question. Consequently, I have no doubts it will be well received by the community, sure to generate considerable interest and further experiments. I also believe the underlying data, once made public, will stimulate additional contributions of equal importance which is why I cannot but recommend the article for publication with only minor revisions.

-pjk

Cambridge

Specific Comments:

1) Re: Fig1. There is clearly a modern sample bias in terms of both annual sample size and series length. How can one be sure this bias is not affecting one's interpretation of the reconstruction? Would it not be possible to depopulate these numbers to a level more consistent with both the mean sample depth and tree age of the entire collection? The explosion of sample size in the modern period, and

the low sample size in the early period are just too outstanding not to be of concern.

2) Re: Extended Data Fig. 5. I can see four lines in the top-right panel as described in the caption, but there are only two lines in the middle and bottom-right panels. Is this intentional?

3) Line 133 RE: Unprecedented heating rate. Would not "Unprecedented warming" be simpler as well as impactful? Related to this within lines 141-143, I was surprised not to see a reference to Esper et al., 2012, Nat. Climate Change?

4) Extended Data Fig. 2 | RCS Detrending. What are the parameters used to define "mean growth rates"? I believe it would be informative to tell the reader, in slightly more detail, how these partitions (growth classes) are achieved. Presumably there are up to four different measures by which growth rates are assessed, what are they? Is the age/length of the series considered at the same time? If a reference can be applied here, then please add it.

a) In the same figure (panel b) suggests the young trees are also the fast growing trees. If the majority of samples in the modern period come from this cohort, could that not create a bias?

5) Figure 3 | Recent unprecedented June-July "heating" - suggest "warmth".

a) All four panels in this figure are missing their letter index; a-c.

6) Extended Data Fig. 4 | Chronology confidence.

a) Consider adding the Sub-Sample Signal Strength (SSS), possibly in lieu of EPS. Given the expected importance of this chronology I believe the SSS is more useful to describe the degree of common signal attributable to replication, particularly during the early portion of the chronology which is influencing the calculation of the overall multi-millennial trend described in Fig. 2 (see: Wigley et al 1984, Cook ER, Kairiukstis LA. 1990, Cook and Pederson 2011, Buras 2017)

b) Remove the minus (-) signs before years BP. Before Present suggests sometime in the past. I would apply this rule to all figures in the manuscript where years (BP) are used as an axis index.

7) Extended data Fig. 6. | Calibration-verification stats of the 3 recons.

a) Are there some values missing from this table (e.g., measures of persistence in residuals, %Variance loss, and the Rvs2 value). What is the Rvs2? If this is not a common statistic in this application should it then be defined somewhere in the text? I presume Rvs2 describes the degree by which the variance in the instrumental and reconstructed values are similar? Maybe a better abbreviation would be EVE

General Comments:

1) In the title, please change "last" to "past" – let's hope these are not the last seven millennia and we will survive the current without apocalyptic consequence. I would apply the same rule to all other similar usages (e.g., Page 6; line 166).

2) Lines 64-66. This sentence could use some polishing. I suggest present tense eg., "Though these records preserve valuable information on climate variability for periods exceeding 2000 years, they retain virtually noshorter than 300 years."

3) Line 88. Change first instance of "of" to "from".

4) Line 96. I would not suggest using the word "divergent" in this instance. "Divergent/divergence in dendrochronology has taken on a new meaning of its own which may in fact be applicable to this study later in the climate calibration phase, but not here.

5) Line 98. Change, "out of inadvertent" to "from the unavoidable".

- 6) Line 130. Here the use of "divergent" is appropriate.
- 7) Lines 171-172. Change order, "exceeds clearly" to "clearly exceeds".

Point-by-point response letter to the “Current Siberian heating is unprecedented during the past seven millennia” manuscript submitted to *Nature Communication*

The reviewer’s comments are displayed in black while our replies are written in red.

Reviewer #1 (Remarks to the Author):

The authors provided well-replicated, high-quality 7,638 years-long tree-ring width chronology at Yamal Peninsula that is one of the northernmost forested region in the World. The chronology is very sensitive to June-July temperature variations. This annually-resolved data set is an impressive paleoclimatic archive that is used to assess the rates and scale of the modern temperature changes in the Arctic in comparison with those over the mid and late Holocene. The chronology itself is the main achievement of this work and this archive will be certainly used in the future by hundreds of researches. The authors identified a weak (0.6C) long-term negative linear trend in the June-July temperature over the entire period covered by their data and claim that this trend was reversed in the industrial era. They consider the current JJ temperature as unprecedented in comparison with the pre-industrial time (from the 8th millennium to 1850s AD) in many respects, namely in term of the number of positive extremes and the degree of current warming.

Thank you very much for this supporting review. We think that we could address your critical comments

1. Long-term trends

It is difficult to compare the 8 ka trend with the one a century-long. Looking at the Fig. 2 the reverse is not obvious, probably due to the scale of the graph.

We agree with the referee that the superposition of several time scales (i.e. interannual, centennial) with the trends as well as the use of gradient colors to highlight the uncertainties blurred the readability of the figure and prevented proper visualization of the recent warming. As a consequence, we modified and simplified Fig 2 (now Fig 3) by increasing the height of the y-axis, simplifying the color scheme and by smoothing the annual time series with a 200-yr spline on a separate panel. The latter now helps to better evidence the amplitude of recent warming. To increase the readability of the 7k Yamal reconstruction, we also provide four additional plots covering 2000-yr periods each in Fig S6. Based on the referee’s comment, we also decided to change the sentences referring to the 8ka trend reversal in lines L29 and L177. The sentences now read as:

L29 “We demonstrate that the recent anthropogenic warming **interrupted** a multimillennial cooling trend^{11,12}”

L177 “Yet, the orbital cooling trend at Yamal Peninsula **was halted** abruptly by pronounced industrial-era warming (1850–2019), leading to an increase of JJ air temperatures without any parallel for comparable time windows (i.e. 170 years) over the mid- to late-Holocene (Fig 4a).”

In addition, one can find – at least at the graph - other cases when the long-term trend was reversed for several decades.

As above, we agree with the referee. Fig 3 (now Fig 4) shows that some other, older periods may have exhibited a warming rate similar to the one reconstructed for the most recent past. We modified the Figure such that the periods with comparable or higher rates than those observed recently are now better visible and clearly labeled in the figure. As a complement to this figure, which shows warming rate vs mean temperature, we produced a new figure (Fig S8) displaying both variables separately and chronologically. Both figures confirm that the periods 1850-2019 (Fig 4a) and 1920-2019 (Fig 4b) are by far the warmest of the last 7ka.

While there are no doubts that the inter-annual variations are well preserved in the chronology the longer-term variability in tree ring records is often questionable. The authors used the multi-curve signal-free Regional Curve Standardization (RCS) in order to preserve as much as possible this low-frequency signal. Moreover, they used two growth-rate curves for fast- and slow-growing trees. This approach is correct, and nowadays I do not know a better way to deal with this problem. To secure their conclusions they also compare their records with the trend from the lower resolution proxies and find them to be similar (0.9 versus 0.6 C). The authors admit that “This is in contrast to some other TRW records that do not show multi-millennial trends”. I would suggest to go into details here, to give appropriate references and explore potential differences in approaches and provide more solid evidence that this trend is not a statistical artefact.

We thank the referee for this comment. Indeed, the original sentence was too vague and confusing for the reader. The original sentence specifically referred to Klippel et al. 2020 (ref. 39) who performed a detailed analysis of pre-industrial long-term trends using 67 chronologies from the Northern Hemisphere extending back to 1 CE. Using different detrending methods (negative exponential, regional curve standardization, as well as signal-free regional curve standardization), the authors concluded that millennial-scale cooling over pre-industrial periods were largely missing in their dataset. One reason potentially explaining the lack of cooling was suggested directly by the authors (Klippel et al., 2020): *"RCS is best applied to large datasets with a homogenous age-structure through time to guarantee a proper representation of the population growth curve used to detrend the data (Esper et al., 2003), and most tree-ring measurements in the 2k database do not satisfy this criterion"*. We agree with this statement and hypothesize (L 165-167) that the multi millennial cooling trend detected in the Yamal7k chronology could be explained by ascribing *"this ability to the high replication and even distribution of samples over time in Yamal7k that facilitate the application of an RCS standardization aimed at preserving long-term trends³⁹"*. We also note that the Yamal7k is not the only reconstruction showing a multimillennial trend. The 5680-year chronology developed by Lara et al (2020) for southern America also shows a Holocene trend attributed to astronomical forcing.

Reference

- Klippel, L., St George, S., Buntgen, U., Krusic, P. J. & Esper, J. 2020. Differing pre-industrial cooling trends between tree rings and lower-resolution temperature proxies. *Clim Past* 16, 729-742, doi:10.5194/cp-16-729-2020
- Esper, J., Cook, E. R., Krusic, P. J., Peters, K., and Schweingruber, F. H. 2003. Tests of the RCS method for preserving low-frequency variability in long tree-ring chronologies, *Tree-Ring Res.*, 59, 81–98.
- Lara, A., Villalba, R., Urrutia-Jalabert, R., et al. 2020. +A 5680-year tree-ring temperature record for southern South America. *Quaternary Science Reviews* 448, doi.org/10.1016/j.quascirev.2019.106087

and provide more solid evidence that this trend is not a statistical artefact.

The long-term cooling trend might indeed be influenced by the tails of the linear regression. To verify that the calculated negative trend of 0.08112 °C kyr⁻¹ over the period -5618BCE-1850CE was not biased by its tails, we recalculated the slope after varying the start and the end of the period by up to 200 years with steps of 50 years (see Table R1). The obtained slopes ranged between 0.08426 and 0.07484 °C kyr⁻¹ with a median of 0.07999 °C kyr⁻¹, indicating the solidity of the cooling trend.

Table R1 | Long-term cooling trend as a function of start and end dates of the considered period

Start [BCE]	End [CE]	Slope [°C kyr ⁻¹]
-5618	1850	-0.08112
-5568	1850	-0.07983
-5518	1850	-0.08370
-5468	1850	-0.08416
-5418	1850	-0.08207
-5618	1800	-0.07622
-5568	1800	-0.07484
-5518	1800	-0.07870
-5468	1800	-0.07909
-5418	1800	-0.07691
-5618	1750	-0.08114
-5568	1750	-0.07981
-5518	1750	-0.08378
-5468	1750	-0.08426
-5418	1750	-0.08212
-5618	1700	-0.07885
-5568	1700	-0.07746
-5518	1700	-0.08147
-5468	1700	-0.08191
-5418	1700	-0.07971
-5618	1650	-0.07910
-5568	1650	-0.07770
-5518	1650	-0.08176
-5468	1650	-0.08222
-5418	1650	-0.07998

Do I understand correctly that the mean temperature at the Fig. 2 (white dashed line for the area across 60–90°N) is up to 0.5 C higher than the mean in Yamal that is located at 67 N, i.e. at the south of the area?

We thank the reviewer for spotting this issue. We indeed forgot to scale Yamal7K to the Kaufman et al. (2020) reconstruction. Temperature anomalies from the Kaufman et al. (2020) anomalies are expressed relative to the

1800-1900 period. To fix the issue raised by the referee, our reconstruction is now plotted on two axes: the left y-axis represents absolute JJ temperature while the right y-axis represents Yamal7K temperature anomalies computed with the same reference period (1800-1900) as Kaufman et al. (2020) (Fig 3c).

Reference

- Kaufman, D. et al. 2020. A global database of Holocene paleotemperature records. *Sci Data* 7, doi:10.1038/s41597-020-0445-3

2. Current warming

The tree-ring-climate model seems to be very solid – the meteorological records are long, the correlation is high. However, at the Fig 2 only very few last years look anomalous especially when averaged, the unsmoothed values do not look unusual. Could it not be partly just a statistical edge effect of the smoothing at the end of the series? When I look at the Fig. 1 in Mann, 2021 (Mann M. E. Beyond the hockey stick: Climate lessons from the Common Era //Proceedings of the National Academy of Sciences. – 2021. – T. 118. – №. 39) I see a clear picture of the unusual warming; when I compare it with the Fig. 2 in this paper I begin to hesitate. Probably it is a matter of representation of the results, but it might be useful to think about the strengthening of the visual message.

The issue raised by the referee is relevant. The apparent absence of extreme warm years over the last decades is a matter of representation and can be explained by (i) the condensed x-axis due to the 7000 year-chronology and (ii) the 100-yr spline fitting which partly masks extreme values. To increase the visibility of the most recent years, we modified and simplified Fig 2 (now Fig 3) by increasing the height of the y-axis, simplifying the color scheme and by smoothing the annual time series with a 200-yr spline on a separated panel. In addition, we also provide four new plots splitting the reconstruction into segments of 2000 years (Fig S6). Fig 4 and Fig S8 further demonstrate the unusual frequency of extreme warm summers over the last decades. Is it however important to highlight here that our analysis (was Fig. 3, now Fig. 4 showing that the amount and rate of warming are unprecedented in the modern era) is based on means and trends in the unsmoothed data and is not, therefore, an artefact of smoothing to the end of the series.

The sample population of 2000-2019 AD is different from those of the other millennia – it is larger and the trees are younger.

There is indeed a higher number of trees available for the period 2000-2019 AD compared to previous millennia. (Fig 2c). Yet the mean ring age is only slightly different, being 99 and 87 years over the entire chronology and over the 1920-2019 periods, respectively.

We also tested for possible biases related to increased sample size over the recent period. To this end, we adjusted sample depths for the recent decades to a level that is consistent with the sample depth of the entire time period. Our results presented in Fig S7 do not suggest differences between the original and adjusted chronologies, therefore confirming the robustness of our results.

At the ED Fig. 2 it is shown that the extreme growth at the end of the curve is rather sensitive to the way you choose for standardization.

This problem is indeed well-known by the tree-ring community. Several studies (e.g. Briffa and Melvin, 2010) have shown that the use of a single RCS curve may not be relevant for trees with widely varying growth rates and that this could result in biases at the end of the chronologies. In order to alleviate this bias, following the suggestion of Briffa and Melvin (2010), we standardized the Yamal tree-ring measurements using two to four regional curves (RCs). The 2-curve SF-RCS chronology only slightly differs from the chronologies obtained with 3 or 4 curves (Fig S1d). We therefore decided to perform the reconstruction using the most parsimonious approach based on 2 curves SF-RCs.

Reference

- Briffa K.R., Melvin T.M., 2010. A Closer Look at Regional Curve Standardization of Tree-Ring Records: Justification of the Need, a Warning of Some Pitfalls, and Suggested Improvements in Its Application. In: Hughes, M., Swetnam, T., Diaz, H. (eds) *Dendroclimatology. Developments in Paleoenvironmental Research*, vol 11. Springer, Dordrecht. https://doi.org/10.1007/978-1-4020-5725-0_5

The ED Fig. 2 shows that there are substantial differences in the growth of trees younger than and older than 100. Can you show a comparison of “the young” trees and “old” trees chronologies with the Salekhard temperature to see how well it is reproduced by the two subsets?

Following the referee’s suggestion, we compared tree-ring chronologies developed from young (<100) and old (>100 yr) trees with JJ temperatures from the Salekhard meteorological station (Fig S1b-c). Over the 1883-2019 period, both chronologies are significantly correlated ($r=0.66$ for Old and $r=0.75$ for Young, $p<0.01$, respectively) with the climate (Fig S1c). Moreover, two subsets are highly correlated with each other (at $r = 0.90$ for both unsmoothed data for the periods 1000-2019 and 1800-2019, respectively; and at $r = 0.80$ or 0.82 smoothed with a 50-yr filter for the periods 1000-2019 and 1800-2019, respectively), confirming that the differences between the young and old tree subsets are not actually that large. These results suggest that the transfer function does not yield substantially different results using old or young trees.

Did the CO₂ fertilization contribute the growth in 20th-21th centuries apart from the climate warming?

This a very interesting point. Several studies indeed documented that CO₂ fertilization had a positive effect on radial growth rates through an increase in water use efficiency mostly for water stressed sites (see e.g. Camarero et al., 2015; Lara et al., 2020, among others). For instance, in southern South America, Lara et al. (2020) developed a 5680-yr long reconstruction from *Fitzroya cupressus* that was adjusted for the effect of CO₂ fertilization. Contrary to Lara et al. (2020) who observed a decoupling between instrumental and TRW chronologies which they attributed to CO₂ fertilization effects over the last decades, no comparable divergence can be found at our study site. These results are in line with free-air CO₂ enrichment (FACE) experiments showing limited influence of CO₂ fertilization on boreal forests (Hickler et al., 2008).

References

- Camarero J.J., Gazol A., Galván J.D., Sangüesa-Barreda G., Gutiérrez E., 2015. Disparate effects of global-change drivers on mountain conifer forests: warming-induced growth enhancement in young trees vs. CO₂ fertilization in old trees from wet sites. *Glob. Chang. Biol.*, 21 (2015), pp. 738-749, [10.1111/gcb.12787](https://doi.org/10.1111/gcb.12787)
- Hickler et al., 2008. CO₂ fertilization in temperate FACE experiments not representative of boreal and tropical forests. *Global Change Biology*, <https://doi.org/10.1111/j.1365-2486.2008.01598.x>
- Lara, A., Villalba, R., Urrutia-Jalabert, R., et al. 2020. +A 5680-year tree-ring temperature record for southern South America. *Quaternary Science Reviews* 448, doi.org/10.1016/j.quascirev.2019.106087

What are three other warmest (high number of extremes) centuries at the Fig 3? Can you zoom up somehow these periods to show that they are different from 1920-2019 AD? At the Fig. 3 very interesting “prominences” pop out of the point cloud. Each “prominence” corresponds to some anomalous period; it is interesting to see what these periods are and how they are formed

In the new Fig 4 we have highlighted the three periods with highest mean JJ temperatures (ending in 1984, 299 and 1070, Fig 4a) and slopes (ending in 2019, -3265 and -2069, Fig 4a). In addition, we also produce a new figure (Fig S8) showing the evolution of mean temperatures and slopes over the last 7k years chronologically.

The problem of the scale of the warming during the Medieval Anomaly is still under discussion. It would be interesting to show more explicitly this comparison.

We compare the Yamal 7K with the ensemble Northern Hemisphere summer temperature reconstruction of Büntgen et al. (2021) and the global annual multi-proxy temperature anomalies of Pages2k consortium (2019) over the period 1 and 2000 CE (also see response to Referee 3). The Medieval Warm Period (950-1250CE) in Yamal was 1.03 °C degree colder than the 1961-1990 reference period, while it was 0.31 °C and 0.25°C warmer for Büntgen et al. (2021) and Pages2k consortium (2019), respectively. Yet, we have concerns regarding the relevance of such a comparison because our reconstruction is representative of Siberia while other reconstructions provide information at the NH and global scales. In addition, the absence of warming at Yamal Peninsula during the Medieval Warm Periods lends support to results from Neukom et al. (2019) demonstrating the lack of spatial and temporal coherence of the Medieval Warm Period at the global scale. For these reasons we decided not to include this comparison in the revised version of the manuscript.

References

- Büntgen, U., Allen, K., Anchukaitis, K.J. et al. The influence of decision-making in tree ring-based climate reconstructions. *Nat Commun* 12, 3411 (2021). <https://doi.org/10.1038/s41467-021-23627-6>
- Neukom, R., Steiger, N., Gómez-Navarro, J.J. et al. No evidence for globally coherent warm and cold periods over the preindustrial Common Era. *Nature* 571, 550–554 (2019). <https://doi.org/10.1038/s41586-019-1401-2>

- PAGES 2k Consortium. Consistent multidecadal variability in global temperature reconstructions and simulations over the Common Era. *Nat. Geosci.* 12, 643–649 (2019). <https://doi.org/10.1038/s41561-019-0400-0>

Fig. 3 shows a very interesting pattern of smaller number of extremes between -2000 to -4000 years. Is it an artefact or a real climatic pattern?

The referee raises a very interesting point here. Attributing the reduced number of extremes between -2000 and -4000 to climate forcing or statistical artefact is a complex task. Indeed, we note that the R_{bar} values and the variance tend to decrease during the period -2000-4000 (Fig S2). One might argue that sample depth could explain the latter trends. Yet, during this period, replication is high, especially between -2400 and -3500 and the EPS systematically remains above the commonly used 0.85 threshold. In addition, we draw the referee's attention to the fact that low R_{bar} values during this period also point to a high frequency of extremes. This is especially the case for the first millennium CE (Fig 4d)

Reviewer #2 (Remarks to the Author):

The manuscript entitled "Current Siberian winter is unprecedented during the last seven Millennia" by Hantemirov et al. presents a novel and multi-millennial tree-ring width chronology from the Siberian Arctic region using three different tree species. The displayed results are based upon a 7,623 year spanning chronology, built from recent as well as historic samples. The displayed results are in general sound, sufficiently well-explained and presented through well—designed figures as well as by underlying statistical analyses. The grammar as well as the overall language is easy to read, while using the respective necessary scientific terms and language. The presented results steadily support the hypotheses/ intended scientific gaps for the region. The used literature is state-of-the art, although the authors could have used more recent multi-millennial work from other authors e.g. for the comparison of their new data with other time series (see my specific comments on that below).

From my perspective, I have not read such a nicely presented and sound manuscript for months.

The authors present their results throughout the manuscript in a clear and sound way. The performed analyses are comprehensible, use state-of-the art dendroclimatological methods and are illustrated in an impressive way. However, some questions arise while reading through the manuscript that justify my review statement as minor revisions.

Thank you very much for this positive and supportive review. We hope that we could address your comments.

Why did the authors use the climate normal period from 1961-1990 instead of using the more recent normal period? At least a comprehensive comparison of both would be helpful to illustrate readers beyond dendroclimatology possible impacts caused by anthropogenic warming for the resulting reconstruction. This also holds true for the main text comparisons (e.g. line 82).

The period from 1961 to 1990 is considered as a standard reference period for long-term climate change assessments (WMO Guidelines on the Calculation of Climate Normals, Nr 1203, 2017). However, following the referee's suggestion, we modified the text in lines L109-L113 by adding the computed climate anomalies relative to the 1991-2020, 1961-1990 and 1850-1900 reference periods.

Reference.

- World Meteorological association, 2017. WMO Guidelines on the Calculation of Climate Normals. ISBN: 978-92-63-11203. <https://public.wmo.int/en/resources/library/wmo-guidelines-calculation-of-climate-normals>

Please describe in more detail the choice of the split sample calibration for your chronology in the methods. This should include some explanation why you neglect or dismiss other state-of-the art methods (e.g. LOOC) or more novel approaches (e.g. K-folded mean) in dendroclimatology. This is especially of importance for the K-folded mean method, since the recent period (1952-2019) is within the most amplified warming period in the used climate record.

The split calibration/verification procedure has been commonly used in a vast majority of dendroclimatic reconstructions over the last decades (see e.g., Fritts, 1976; Büntgen et al., 2021). The choice of this procedure was motivated by the length of the Salekhard meteorological series covering the 1883-2019 period. This long series ensures the robustness of calibration/validation statistics computed over 68-yr long period. Yet following

the referee's recommendation, we also tested the K-fold mean calibration-verification method. We randomly partitioned the meteorological and TRW data into 10 equally-sized sub-data sets (known as folds). Nine of these folds, representing 80% of the data, were used to train the model while the remaining fold (20% of the data) was used for testing. The process was repeated 10 times. The K-fold mean and the more traditional split calibration/verification procedures yield comparable calibration/validation statistics (r , RE, CE) provided in Fig S5 and shown in Fig S4.

Reference

- Büntgen, U., Allen, K., Anchukaitis, K.J. et al. The influence of decision-making in tree ring-based climate reconstructions. *Nat Commun* 12, 3411 (2021). <https://doi.org/10.1038/s41467-021-23627-6>
- Fritts HC.1976. *Tree-rings and Climate*. Academic Press, New-York, 568 pp.

In addition to this, have you checked the usage of gridded re-analysis products to further underline your results? How about analyses investigating the regional representativity of your reconstruction?

In order to highlight the spatial representativity of our study, we computed the spatial correlation between the Yamal chronology and the CRU TS4 gridded dataset (0.5 x 0.5° lat/long resolution) (Fig S3c). The map shows that the highest correlations between the Yamal 7k chronology and June-July temperatures over the 1901-2019 period are observed at the vicinity of Yamal Peninsula and exceed 0.6 in a region with latitudes ranging between 60 and 70° N and longitude comprised between 65 and 85° East.

This could stimulate the discussion part and the outreach points on the overall usage of your dataset e.g. compared to datasets from High Asia (e.g. by the tree-ring oxygen isotope based dataset presented in Bao et al, *PNAS* 2021). The overall comparison with other regional / northern hemispheric temperature proxy records represented in the ms is astonishingly low and deserves more focus. You mentioned of course the lack of available high-res (TRW) data, but there are records from other proxy archives and / or from other Arctic locations in the Northern hemisphere that are worth for comparing. This includes incorporating a graph illustrating common periods or contrasting trends also used for the overall discussion (e.g. with the recent NH recon comparison paper by Buntgen et.al. 2021). How about using additionally modelled /simulated GCM paleodata for comparison?

Over the last decade, several studies (Bartlein et al. 2011; Marcott et al. 2013; Sundqvist et al. 2014; Marsizek et al. 2018; Kaufmann et al 2020) have reconstructed Holocene temperature across the Arctic regions. As previously stated in our manuscript, each of these studies shows a lack of highly-resolved proxies in Western Siberia. The most recent study which relies upon the most exhaustive database confirms that not a single proxy record is available for comparison with Yamal (see Fig 6 in Kaufmann et al. 2020).

We agree with the referee that there might be potential teleconnections between Yamal and Central Asia that could be explored in more detail, yet we believe that such an analysis would be out of scope within this paper focusing on the unprecedented character of recent global warming in Siberia. Nevertheless, we explored the linkage with Asian precipitation reconstructions and could not find any significant correlation between our annual tree-ring chronology and the isotope chronology from Yang et al. 2021 ($r = 0.29$ on 100 yr splines, and $r = -0.01$ between the 100 yrs spline detrended residual chronologies, see Fig R1).

Fig R1 | Comparison between the Yamal7k Temperature reconstruction and Yang et al. 2021 precipitation reconstruction over the past 7000 years. Due to irregular resolution in Yang et al. 2021, the Yamal7k temperature was matched to the Yang et al. resolution before fitting a 100 yrs cubic smoothing spline functions.

Similarly, we compared our Yamal 7k reconstruction with the ensemble Northern Hemisphere summer temperature reconstruction of Büntgen et al. (2021) and the global annual multi-proxy temperature anomalies of Pages2k consortium (2019) (Fig R2). Over the period 1-2000 CE, the correlations between the 100 years splined Yamal 7K, Büntgen et al (2021) and PAGES2k are 0.59 and 0.32, respectively. Although we observe similar temperature fluctuations for some periods (for instance during the cold 1800-1820 CE decade coinciding with the Dalton solar minimum or during the 200-300 CE warm period), we note that the three reconstructions are not fully independent from one another as the Yamal chronology is part of the Büntgen et al. (2021) and PAGES 2k (2019) networks. In addition, we also wonder whether comparing a regional chronology with a NH and global reconstructions is relevant.

The Yamal 7K reconstruction could also be compared with output from recent transient Holocene simulations available from e.g., Braconnot et al. (2019) or Tian et al. (2022). Yet, we would like to emphasize that such a comparison would go beyond the scope of this paper which aims to put the current warming in Western Siberia in the Holocene perspective.

Fig R2 | Comparison with Büntgen et al. 2021 and Pages2k 2019 temperature reconstruction. Panel a compares the Yamal7k JJ temperature reconstruction (gray line = annual value, green band = reconstruction error, black thick line = 100-year cubic spline fitting) with the mean of the 15 ensemble Northern-Hemisphere tree-ring based summer temperature anomalies of the past 2 millennia (Büntgen et al. 2021) (light red line and red line = 15 reconstructions and its mean, thick red line = 100-year cubic spline fitting). Panel b compares the Yamal7k JJ temperature reconstruction (gray line = annual value, green band = reconstruction error, black thick line = 100-year cubic spline fitting) with the ensemble global multiproxy-based annual (April-March) temperature anomalies of the past 2 millennia (Pages2k consortium, 2019) (light blue lines = 2.5 and 97.5 percentile, blue line = full ensemble median, thick blue line = 100-year cubic spline fitting). The temperature anomalies refer to the 1961-1990 period and are shifted to match the temperature level of the Yamal7k temperature reconstruction for the same period. The period 1961-1990 CE is indicated by the vertical lines.

Reference

- Bartlein, P.J., Harrison, S.P., Brewer, S. et al. Pollen-based continental climate reconstructions at 6 and 21 ka: a global synthesis. *Clim Dyn* 37, 775–802 (2011). <https://doi.org/10.1007/s00382-010-0904-1>
- Braconnot, P., Zhu, D., Marti, O., and Servonnat, J.: Strengths and challenges for transient Mid- to Late Holocene simulations with dynamical vegetation, *Clim. Past*, 15, 997–1024, <https://doi.org/10.5194/cp-15-997-2019>, 2019.
- Büntgen, U., Allen, K., Anchukaitis, K.J. et al. The influence of decision-making in tree ring-based climate reconstructions. *Nat Commun* 12, 3411 (2021). <https://doi.org/10.1038/s41467-021-23627-6>
- Kaufman, D., McKay, N., Routson, C. et al. Holocene global mean surface temperature, a multi-method reconstruction approach. *Sci Data* 7, 201 (2020). <https://doi.org/10.1038/s41597-020-0530-7>
- Marcott, S. A., Shakun, J. D., Clark, P. U. et al. A reconstruction of regional and global temperature for the past 11,300 years. *Science* 339, 1198–1201 (2013)
- Marsicek, J., Shuman, B., Bartlein, P. et al. Reconciling divergent trends and millennial variations in Holocene temperatures. *Nature* 554, 92–96 (2018). <https://doi.org/10.1038/nature25464>
- PAGES 2k Consortium. Consistent multidecadal variability in global temperature reconstructions and simulations over the Common Era. *Nat. Geosci.* 12, 643–649 (2019). <https://doi.org/10.1038/s41561-019-0400-0>
- Sundqvist, H. S., Kaufman, D. S., McKay, N. P., Balascio, N. L., Briner, J. P., Cwynar, L. C., Sejrup, H. P., Seppä, H., Subetto, D. A., Andrews, J. T., Axford, Y., Bakke, J., Birks, H. J. B., Brooks, S. J., de Vernal, A., Jennings, A. E., Ljungqvist, F. C., Rühland, K. M., Saenger, C., Smol, J. P., and Viau, A. E.: Arctic Holocene proxy climate database – new approaches to assessing geochronological accuracy and encoding climate variables, *Clim. Past*, 10, 1605–1631, <https://doi.org/10.5194/cp-10-1605-2014>, 2014.
- Tian, Z., Jiang, D., Zhang, R., and Su, B.: Transient climate simulations of the Holocene (version 1) – experimental design and boundary conditions, *Geosci. Model Dev. Discuss.* [preprint], <https://doi.org/10.5194/gmd-2022-49>, in review, 2022

More specific comments:

Please insert in Fig 2 the running sample replication and the EPS as individual layers. It is only readable if using a magnifier when the individual series shifts from grey to purple... This clearly enhances the overall readability/information on time series statistics. The individual series given in grey can be then e.g. in a more light grey.

The figure has been modified according to the referee's suggestion to highlight more clearly the periods with EPS values below the 0.85 threshold. The running sample replication is shown in Fig 3 and Fig S2.

Extended data Fig. 1.: the periods with insufficient sample replication should be marked more prominent. Within the used coloring scheme, it is not easy to recognize that. In addition, you stated the "Sentinel scene is used for giving an overview of sample replication. Well, I have never heard that a satellite image can be used to get information on sample replication, I think you meant that you used it for illustrate a) location of samples in your study area and b) additionally give information on the respective location of long/short series. However, the overall readability of this in your graph a) is weak. Why did you not use simply different age classes instead of a floating classification? It is quite hard to read anything in the color floating legend where e.g. the more older samples are from (tiny graph inlay for the amount of information you present here).

The figure and caption (now as Figure 2 in the main text) have been modified according to the referee's suggestion. We also replaced the continuous color scale used on the map tree-age across space by a discrete color scale to increase readability of the figure. Periods with low sample replication, initially displayed with a grey color in Fig 2c, are now better highlighted in Fig. 3 and Fig S2 with red colored surface under the sample count line.

Reviewer #3 (Remarks to the Author):

Peer Review

RE: Current Siberian heating is unprecedented during the last seven millennia

Authors: Rashit M. Hantemirov et al. 2021

Nature Communications manuscript: NCOMMS-21-43050-T

Preface:

To the editor, this is not an anonymous review. Please share the entirety with the authors.

Summary:

I am impressed. This work clearly represents a tremendous effort spanning many years and involving many participants on multiple levels. I am very pleased to see it finally coming together in a such a well presented and thorough manner. It had been a long time coming, a chronology of this length from a region of significant paleoclimate importance. As mentioned, the manuscript is well constructed, written, and cogent. I have found absolutely no glaring technical issues regarding the analyses involved to question. Consequently, I have no doubts it will be well received by the community, sure to generate considerable interest and further experiments. I also believe the underlying data, once made public, will stimulate additional contributions of equal importance which is why I cannot but recommend the article for publication with only minor revisions.

-pjk

Cambridge

Thank you very much for this supportive review. We hope that we could address all your comments.

Specific Comments:

1) Re: Fig1. There is clearly a modern sample bias in terms of both annual sample size and series length. How can one be sure this bias is not affecting one's interpretation of the reconstruction? Would it not be possible to depopulate these numbers to a level more consistent with both the mean sample depth and tree age of the entire collection? The explosion of sample size in the modern period, and the low sample size in the early period are just too outstanding not to be of concern.

This is a very relevant comment. There is indeed a higher number of trees available for the last century compared to previous centuries (Fig 2c). Yet, the mean ring age is only slightly different, being on average 86 and 79 over the entire chronology and the 1850-2019 periods, respectively.

Following the referee's suggestion, we adjusted sample depth for the last century to a level that is consistent with the entire collection. Results presented in Fig S7 do not suggest any significant differences between the original and adjusted chronologies therefore confirming the absence of possible inflation of recent chronology values arising out of inadvertent selection of mostly relatively fast-growing trees in recent centuries. This absence of modern bias is mentioned in the manuscript in lines L119-123, L157-160 and L269-272.

2) Re: Extended Data Fig. 5. I can see four lines in the top-right panel as described in the caption, but there are only two lines in the middle and bottom-right panels. Is this intentional?

The top left-panel of Fig S4 compares the instrumental series from the Salekhard meteorological station and estimated from the Yamal chronology using the multiple frequency band approach which combines three High, Medium and Low frequency bands (<15 years, 15–100 years, and >100 years) to reconstruct past temperature variability (Briffa et al., 2013). The three red lines in the top left-panel of Fig S4 represent the high (thick line), medium and low (dotted line) frequency components of the Yamal reconstruction, respectively. The three black lines represent the high (thick line), medium and low (dotted line) frequency components of the instrumental records, respectively. The caption has been updated accordingly.

Reference

- Briffa, K.R., Melvin, T.M., Osborn, T.J., Hantemirov, R.M., Kirilyanov, V.S., Shiyatov, S.G., Esper, J. (2013) Reassessing the evidence for tree-growth and inferred temperature change during the Common Era in Yamalia, northwest Siberia. *Quaternary Science Reviews* 72: 83 – 107

3) Line 133 RE: Unprecedented heating rate. Would not “Unprecedented warming” be simpler as well as impactful?

We agree with the referee. The sentence has been modified accordingly.

Related to this within lines 141-143, I was surprised not to see a reference to Esper et al., 2012, *Nat. Climate Change*?

Thank you for the comment. Indeed, this reference was missing and was added to the revised version of the manuscript at L169.

4) Extended Data Fig.2 |RCS Detrending. What are the parameters used to define “mean growth rates”? I believe it would be informative to tell the reader, in slightly more detail, how these partitions (growth classes) are achieved. Presumably there are up to four different measures by which growth rates are assessed, what are they? Is the age/length of the series considered at the same time? If a reference can be applied here, then please add it.

The method used to estimate mean growth is based on the methodology defined by Melvin (2004), Melvin et al. (2014) and implemented in the CRUST software. The method used to sort trees by “growth rate” is to compare each tree's growth rate with the growth rate of a single RCS curve created using all trees. The relative growth rate is obtained by the ratio between the sum of measurements of a tree (i.e. ~ the radius of the tree) divided by the sum of the RCS curve values created using all trees. Where estimates of missing radius and missing years (rings) to the center of the trees are available, these are used in the creation of the single RCS curve and the estimation of final tree radius and final tree age. Trees are then sorted by “relative” growth rates and roughly equal numbers of trees are used to create each of a series of RCS curves representing increasing growth rate classes. CRUST will only use multiple RCS if there are data from 40 or more trees in each RCS curve. As suggested by the referee, additional details are now included in the methods section (L243-247) and in the caption of Fig S1.

Reference

- Melvin T.M., 2004. Historical Growth Rates and Changing Climatic Sensitivity of Boreal Conifers. PhD thesis, Climatic Research Unit School of Environmental Sciences University of East Anglia, 255p.
- Melvin T. M., Briffa K. R., 2014. CRUST: Software for the implementation of Regional Chronology Standardisation: Part 1. *Signal-Free RCS. Dendrochronologia* 32, 7–20.

a) In the same figure (panel b) suggests the young trees are also the fast growing trees. If the majority of samples in the modern period come from this cohort, could that not create a bias?

Thank you for this very relevant comment. As stated above, the adjustments of sample depth and tree age for the recent period to the levels of previous centuries did not have significant effects on the reconstruction (see Fig S7). In other words, it demonstrates the robustness of our multiple-curve, signal-free RCS detrending

approach which efficiently removes the modern sample bias that could potentially have been induced by the inclusion of a larger number of faster growing young trees over the last century.

5) Figure 3 | Recent unprecedented June-July “heating” -suggest “warmth”.

As suggested, we replaced “heating” with “warmth” (see also reply to point 3 of reviewer 3)

a) All four panels in this figure are missing their letter index; a-c.

The index letters have been added accordingly.

6) Extended Data Fig. 4 | Chronology confidence.

a) Consider adding the Sub-Sample Signal Strength (SSS), possibly in lieu of EPS. Given the expected importance of this chronology I believe the SSS is more useful to describe the degree of common signal attributable to replication, particularly during the early portion of the chronology which is influencing the calculation of the overall multi-millennial trend described in Fig. 2 (see: Wigley et al 1984, Cook ER, Kairiukstis LA. 1990, Cook and Pederson 2011, Buras 2017)

As suggested by the referee, the SSS and the mean age have been added to the Fig S2.

b) Remove the minus (-) signs before years BP. Before Present suggests sometime in the past. I would apply this rule to all figures in the manuscript where years (BP) are used as an axis index.

We agree with the reviewer. All the figures have been adjusted accordingly.

7) Extended data Fig. 6. | Calibration-verification stats of the 3 recons.

a) Are there some values missing from this table (e.g., measures of persistence in residuals, %Variance loss, and the Rvs2 value). What is the Rvs2? If this is not a common statistic in this application should it then be defined somewhere in the text? I presume Rvs2 describes the degree by which the variance in the instrumental and reconstructed values are similar? Maybe a better abbreviation would be EVE

The table has been filled with the Durbin Watson (i.e. an indices of the persistence in residuals), %Variance loss, and the Rvs2 values relative to the early and late calibration period. Moreover, Rvs2 is indeed the Equivalent variance explained, see Figure caption. To avoid confusion, we changed the acronym from Rvs2 to EVE.

General Comments:

1) In the title, please change “last” to “past” – let’s hope these are not the last seven millennia and we will survive the current without apocalyptic consequence. I would apply the same rule to all other similar usages (e.g., Page 6; line 166).

We changed “last” to “past” in the title and everywhere in the text, when referring to the time period covered by the chronology

2) Lines 64-66. This sentence could use some polishing. I suggest present tense eg., “Though these records preserve valuable information on climate variability for periods exceeding 2000 years, they retain virtually noshorter than 300 years.”

The sentence has been changed as suggested

3) Line 88. Change first instance of “of” to “from”.

The sentence has been changed as suggested

4) Line 96. I would not suggest using the word “divergent” in this instance. “Divergent/divergence in dendrochronology has taken on a new meaning of its own which may in fact be applicable to this study later in the climate calibration phase, but not here.

The sentence has been changed and we used the term “different”

5) Line 98. Change, “out of inadvertent” to “from the unavoidable”.

We believe that the suggested change slightly but inappropriately modifies the meaning. The use of “unavoidable” implies the impossibility for a fair sampling selection in modern time, which we consider to be too absolute. We therefore kept the original text.

6) Line 130. Here the use of “divergent” is appropriate.

The wording has been kept

7) Lines 171-172. Change order, “exceeds clearly” to “clearly exceeds”.

The sentence has been changed as suggested

Reviewers' Comments:

Reviewer #1:

Remarks to the Author:

The authors provided detailed and comprehensive comments to all my questions and changed some figures to highlight their important findings. In my opinion the manuscript is ready for publication. It was a very long way of search, collection, and analyses, and I would like to congratulate the authors with this excellent scientific product that they submitted to a very appropriate high-ranking Journal.

Reviewer #2:

Remarks to the Author:

The revised manuscript entitled "Current Siberian heating is unprecedented during the last seven Millennia" by Hantemirov et al. presents a novel and multi-millennial tree-ring width chronology from the Siberian Arctic region using three different tree species. The displayed results are based upon a 7,623 year spanning chronology, built from recent as well as historic samples. The displayed results of this revised version are fully sound, sufficiently well-explained and presented through well-designed figures as well as by underlying statistical analyses.

From my perspective, the authors extensively and thoroughly responded to my few concerns/suggestions and addressed all respective gaps or suggestions for changes in figure formats (as also for the other reviewers). Therefore, I am recommending this paper being accepted by the Editorial office. Congratulations to this impressive work!

Reviewer #3:

Remarks to the Author:

Once again I am pleased to see this research progress well in the review process. As this is the second time I am reviewing the manuscript I have only to review how the authors addressed my, and my fellow referees comment. To that end I am thoroughly satisfied. My only contribution at this point is to apologize for not returning this reply sooner.

Sincerely,

-pjk

Cambridge.